# The Impact of COVID Vaccination on Symptoms of Long COVID: An International Survey of People with Lived Experience of Long COVID

**DOI:** 10.3390/vaccines10050652

**Published:** 2022-04-21

**Authors:** William David Strain, Ondine Sherwood, Amitava Banerjee, Vicky Van der Togt, Lyth Hishmeh, Jeremy Rossman

**Affiliations:** 1Diabetes and Vascular Research Centre, University of Exeter Medical School, Barrack Road, Exeter EX2 5AX, UK; 2Academic Department of Healthcare for Older People, Royal Devon & Exeter Hospital Barrack Road, Exeter EX2 5DW, UK; 3LongCovidSOS, Patient Advocacy Group, Surrey, UK; ondinesh@aol.com (O.S.); lythb7@hotmail.com (L.H.); 4Institute of Health Informatics, University College London Hospitals NHS Trust, London NW1 2DA, UK; ami.banerjee@ucl.ac.uk; 5Research-Aid Networks, Chicago, IL 60605, USA; vickyvdtogt@gmail.com; 6School of Biosciences, University of Kent, Kent CT2 7NJ, UK; jeremy@researchaidnetworks.org

**Keywords:** long COVID, vaccination, survey

## Abstract

Long COVID is a multi-system syndrome following SARS-CoV-2 infection with persistent symptoms of at least 4 weeks, and frequently for several months. It has been suggested that there may be an autoimmune component. There has been an understandable caution amongst some people experiencing long COVID that, by boosting their immune response, a COVID vaccine may exacerbate their symptoms. We aimed to survey people living with long COVID, evaluating the impact of their first COVID vaccination on their symptoms. Methods: Patients with long COVID were invited to complete a web-based questionnaire through postings on social media and direct mailing from support groups. Basic demographics, range and severity of long COVID symptoms, before and after their vaccine, were surveyed. Results: 900 people participated in the questionnaire, of whom 45 had pre-existing myalgic encephalomyelitis or chronic fatigue syndrome (ME/CFS) but no evidence of COVID infection, and a further 43 did not complete the survey in full. The demographics and symptomology of the remaining 812 people were similar to those recorded by the UK Office of National Statistics. Following vaccination, 57.9% of participants reported improvements in symptoms, 17.9% reported deterioration and the remainder no change. There was considerable individual variation in responses. Larger improvements in symptom severity scores were seen in those receiving the mRNA vaccines compared to adenoviral vector vaccines. Conclusions: Our survey suggests COVID-19 vaccination may improve long COVID patients, on average. The observational nature of the survey limits drawing direct causal inference, but requires validation with a randomised controlled trial.

## 1. Article Summary

### Strengths & Limitations

This manuscript addresses a common concern held by people living with long COVID when considering vaccination.

We performed a survey of >800 people with long COVID to describe their response to vaccination, demonstrating that the majority of participants improve after their vaccination, albeit this is temporary in most.

The cross-sectional observational nature of this study is a significant limitation to proposing a mechanistic intervention, however it should be reassuring for those with long COVID having doubts about receiving their vaccine.

## 2. Introduction

The persistence of symptoms and disruption to health following viral infections have been described in previous epidemics, such as coronaviruses, severe acute respiratory syndrome (SARS-CoV) and Middle East respiratory syndrome [1], Ebola [2] and chikungunya [3]. At the time of writing, the SARS-CoV-2 pandemic has infected 473.3 m people worldwide [4] and has been shown to cause long-term sequalae in not only people hospitalised with severe disease but also those who managed their acute disease in community settings. The first UK estimate of self-reported long COVID symptoms produced by the Office of National Statistics (ONS) suggested 21% (95% CI 19.9–22.1%) of individuals had significant limiting symptoms more than 5 weeks after their illness [5]. This declined to 9.9% after 12 weeks. As the methodology of case definition has been refined in conjunction with the scientific subgroup of the English National Health Service (NHS) Long COVID Taskforce, this estimate has now been modified to 14.6% of the unvaccinated population self-reporting persistent symptoms lasting beyond 12 weeks, compared to 9.5% of the fully vaccinated population. Other studies estimate greater prevalence. The first cohort of hospitalised individuals that underwent detailed observation were the initial Wuhan Cohort in China [6]. This reported 76% of the population had residual symptoms at 6 months. For many, this was the post intensive therapy unit (ITU) syndrome, however a considerable number had new symptoms, including, but not limited to, new chronic kidney disease, that could potentially require lifelong treatment, fatigue, shortness of breath, and diarrhoea [7]. A Norwegian study performed a detailed evaluation of 312 patients, recruited during the first wave. Cases were followed up for 6 months; some were milder (247 home-isolated) and others more severe (65 were hospitalized) [8]. The very detailed follow-up of this population demonstrated rates of on-going symptoms at 6 months much higher than the current ONS self-reporting estimates in the UK, estimated at 61% across the population. This was highest at 81% in the hospitalised population, reducing to 55% for those that were able to home isolate. Of concern, over half of the population from 15–30 years reported ongoing symptoms beyond 6 months. People experiencing these symptoms coined the term, and recognised long COVID, before researchers and clinicians. They have mobilised the global community around long COVID yet the debate around impact of vaccines has not taken patient views and experience into account. In this study, we address this gap.

The term “long COVID” remains the patient-preferred name for this condition [9], although the National Institute for Health Excellence (NICE) uses the term “Post COVID-19 Syndrome”, defined as: “Signs and symptoms that develop during or after an infection consistent with COVID-19, continue for more than 12 weeks and are not explained by an alternative diagnosis” [10]. The World Health Organization (WHO) further refined this definition by a Delphi consensus to a condition which “…occurs in individuals with a history of probable or confirmed SARS-CoV-2 infection, usually 3 months from the onset of COVID-19 with symptoms that last for at least 2 months and cannot be explained by an alternative diagnosis [11].“ Survey data from online support groups describe wide-ranging symptoms impacting different body systems, sometimes relapsing and remitting in nature and persisting for many months [12,13]. Indeed, a recent meta-analysis identified over 50 long-term symptoms [14] affecting up to 80% of people in the post-acute viral phase. These range from fatigue, headaches and brain fog affecting 58%, 44% and 27% of people, respectively, through to dizziness, stroke and mood disorders, each affecting less than 5%. The impact on daily activities and ability to work can be significant [15]. The mechanism underlying these residual symptoms is not yet established [16]; however, hypotheses include a post-viral syndrome with similarities to Chronic Fatigue Syndrome (CFS) or Myalgic Encephalomyelitis (ME), unregulated or auto-immune response, viral persistence not easily identified by current testing methods [17], organ damage [18] and microvascular changes [19].

At present, there are no established treatments for long COVID [10]. More than 80 long COVID clinics have been established by NHS England; however, these are termed ‘assessment centres’ reflecting the lack of evidence-based treatment available. 

During the initial rollout of vaccines to protect against SARS-CoV-2 in early 2021, individuals expressed concern to patient support groups (via personal communication with OS, VvdT and LH) that vaccination may exacerbate any autoimmune disorder. In order to address this, we surveyed members across international long COVID support groups to determine the impact of vaccination on their symptoms. 

## 3. Methods

The survey was co-designed and co-implemented between researchers at the University of Exeter Medical School, University of Kent, LongCovidSOS and the ZeroCovid Alliance. Respondents were invited to participate through social media posted online by the LongCovidSOS patient advocacy group on their website and promoted on Twitter, in the international Body Politic COVID-19 Support Group and in several UK-based and international long COVID Facebook groups (Israel, Russia, India, South Africa). An invitation to participate was also sent to the LongCovidSOS subscriber email list. Participants were encouraged to wait until a week after vaccination before completing the survey to avoid the results being unduly impacted by adverse immediate reactions to the vaccine. This produced a cross-sectional convenience non-probability sample of people with lived experience. The survey was open to those with current or recent (at the time of vaccination) symptoms of long COVID, with a diagnosis of COVID-19 based on PCR/antibody testing, symptoms and contact with a proven case or symptoms alone. Due to the lack of availability of testing early in the COVID-19 pandemic, and in keeping with NICE guidance, a serological confirmation was not required in order to be regarded as long COVID [10]. 

Data was gathered via an online Google form between 16 March 2021 and 5 April 2021 inclusive. It was made clear to participants that by completing the survey they were consenting to their anonymised data being used for this research project. This survey of a patient group’s contacts, independent of the NHS, did not require HRA/Ethics approval [20].

## 4. Questionnaire Structure

Participants were asked to provide information on their COVID-19 testing status and severity of initial infection (asymptomatic, mild symptoms, moderate, hospitalisation, ICU). They were invited to provide a date of infection and to choose from five time periods in case the date was not known. Information was requested regarding chronic conditions as well as demographic data (age, biological sex and ethnic group).

The survey asked respondents to assess the severity of their long COVID symptoms prior to their vaccination on a scale of 1 to 10, with 1 = very mild and 10 = very severe, and an absence of symptoms scored as a zero. Data was gathered on the date and type of each vaccination. We asked participants for an overall assessment of the change in symptoms after the vaccine and to score each symptom for severity on the same scale. This series of questions was repeated for those who had received a second vaccine.

## 5. Analysis

For this survey, no formal power calculations were made. Baseline characteristics are presented without formal statistical analysis. When determining the correlates of each symptom, multivariate regression analysis was performed evaluating the impact of each vaccine on symptoms, adjusted for baseline symptom score, age group (to within 5 years), sex, ethnicity and duration of long COVID symptoms. Mean and 95% CI are presented after adjustment, with a positive number representing an improvement in symptoms, and a negative number suggesting deterioration. Whereas the intrapersonal reproducibility of visual analogue scores is good, the interpersonal agreement is less satisfactory. Therefore, we analysed the individual percentage change in symptom score, rather than the absolute difference in symptom score. Sensitivity analyses were performed considering only respondents who had PCR or antibody confirmed COVID-19 infections, and a further analysis that included those with a confirmed COVD-19 contact in addition to symptoms. In keeping with the recommendations of Cupples [21] and Rothman [22], the measured significance of the variables of interest is reported without adjustment for multiple testing. Where presented, the significance of co-variates within the models is presented only after Bonferonni correction. Statistical significance was considered at *p* < 0.05. Statistical analysis was performed using Stata SE 16.1 (Mac version: Statacorp Ltd., College Station, TX, USA).

## 6. Results

900 people completed the questionnaire, of whom 855 reported either definite or probable COVID infection. The remaining 45 had experienced exacerbations of their pre-existing Chronic Fatigue Syndrome (CFS) or Myalgic Encephalomyelitis (ME), but no clear COVID infection. A further 43 participants did not complete their post-vaccine symptom status, and thus were excluded from the analysis. Thus, 812 individuals were included in the final analysis. The majority of patients (72.4%) had experienced their long COVID symptoms for more than 9 months. Most patients had mild or moderate symptoms, with only around 10% of patients requiring hospitalization for their acute COVID. In accordance with the availability of testing early in the pandemic, only 40% had received a diagnostic confirmatory test, either with a PCR or an antigen test. All individuals had received the first dose of a vaccine, predominantly Astra-Zeneca vaccine, followed in frequency by the Pfizer vaccine and Moderna. (Table 1). The median time gap between vaccination and completing the survey was 9 weeks (range of 1 week to 21 weeks). The demographics of our population were very similar to the ONS report on long COVID, with a predominance of younger females [5]. Fatigue, “brain fog”, myalgia and shortness of breath were the dominant symptoms, also in keeping with the ONS report (Table 2).

In the total population, the number of respondents reporting no or mild symptoms (scoring 0–4 out of 10) was greater for every symptom after vaccination compared to pre-vaccination whereas the number reporting moderate or severe symptoms (any score greater than 4 out of 10) reduced post vaccination (Figure 1). Following vaccination, 57.9% of participants reported an overall improvement in symptoms, broken down into 58%, 56%, and 66% with the AstraZeneca, Pfizer and Moderna vaccine, respectively, whereas 17.9% reported a deterioration of their average symptoms after vaccination (made up of 19%, 18% and 12% for AstraZeneca, Pfizer and Moderna vaccine, respectively), with the rest reporting no difference. Only 3% of individuals reported that all of their symptoms deteriorated, compared to 27.2% reporting only improvement.

Across the whole population, the average symptom score was improved after vaccination, for all symptoms, except fever after the AstraZeneca/Oxford vaccine, where there was no change. The average improvement across all symptoms was 22.6% of the baseline symptom score after the AstraZeneca/Oxford vaccine, 24.4% after the Pfizer/BioNTech vaccine, and 31.0% in recipients of the Moderna (*p* = 0.003 compared to the AZ/Oxford vaccine and *p* = 0.01 compared to Pfizer/BioNTech). As anticipated, the absolute change of symptoms was proportionate to baseline symptoms, such that the worse the original symptoms the greater the potential for change; indeed, for each symptom the association between change from baseline correlated with baseline (*p* < 0.0001) for each symptom (data not shown). When exploring individual symptoms, after adjustment for baseline symptom score, sex, age group and duration of long COVID symptoms, the mRNA Moderna vaccine technology compared favourably with the adenoviral vector AZ/Oxford vaccine for improvements in fatigue (*p* = 0.009), brain fog (*p* = 0.01), myalgia (*p* = 0.006), gastro-intestinal symptoms (*p* = 0.05) and autonomic dysfunction (*p* = 0.004) (Table 3).

For approximately half of the patients (52.3%), the improvement of symptoms had abated by the time they completed the survey, with the median duration of improvement between 14 and 21 days. For those who experienced a decline in symptoms post vaccination, exactly 50% had recovered by the time of the survey, with the median time to improvement between 3–7 days, suggesting the deterioration was a vaccination reaction rather than a true exacerbation of long COVID.

A sensitivity analysis exploring the impact of vaccines on fatigue, brain fog and myalgia in the 328 participants who had their diagnosis confirmed with either a PCR or an antigen test demonstrated numerically similar results as with the full data set. Specifically, fatigue improved by 11.2%, 13.8% and 33.8% for AZ/Oxford, Pfizer and Moderna vaccines, respectively; brain fog by 18.8%, 25.6% and 35.6% and myalgia by 13.3%, 9.2% and 34.9%, respectively. The reduction of power in this smaller cohort rendered none of these changes significant from each other.

Only 130 respondents had received their second dose of vaccine. The second dose was associated with a modest further improvement in symptoms or a maintenance of benefit; however, with such small numbers statistical analysis would not be relevant.

## 7. Discussion

We have performed the largest survey to date of people living with long COVID, demonstrating a reduction in symptoms after vaccinations of approximately almost a quarter. There was significant difference between those who received the Moderna vaccine compared to those who received the AstraZeneca/Oxford vaccine for the key symptoms of fatigue, myalgia and chest pain. The observational nature of these data require validation in a prospective randomised trial; however, the findings of a significant difference that is unaccounted for by patient demographics, prior severity and duration of symptoms, suggests that there is a potential for future studies.

Our results are in keeping with a previous survey of approximately half the size, in which half of respondents saw no change in their symptoms, 32% felt slightly or completely better, compared to 18% who felt worse. In that study, the mRNA vaccines (Pfizer and Moderna) also performed significantly better when compared to modified adenoviral vector vaccine (AstraZeneca). A study involving only 44 previously hospitalised long COVID patients found a small overall improvement in symptoms compared to controls after vaccination, but did not identify any difference in response between vaccine types [23].

Here we see that the impact of vaccination on long COVID symptoms is highly variable, similar to previous work. Vaccination was associated with improvements in long COVID symptoms for over half of all respondents; however, between 1 in 7 and 1 in 5 reported a worsening of symptoms. No respondent characteristics reported in this survey were identified that could predict vaccine effect on symptomology.

The survey also explored the differential effect of the vaccination with mRNA technology (Pfizer and Moderna) compared to adenoviral-vectored vaccine (AstraZeneca/Oxford). There appeared to be an advantage for those receiving the mRNA vaccinations compared to the adenoviral vector vaccines. This was most notable for the classic triad of fatigue, brain fog and myalgia.

Interestingly, there are a growing number of reports suggesting that long COVID is less frequent in vaccinated populations [24,25,26]. Given long COVID is based on self-reported symptoms, this has raised the possibility that vaccines are ameliorating the features, and therefore are reducing the presentation of the condition rather than addressing the underlying causes.

### Hypothesised Mechanisms

The improvement seen after vaccination was contrary to the concern expressed by many sufferers that long COVID was an autoimmune condition [27]. Had this been the case, we would have expected the condition to deteriorate rather than improve. Since the survey was conducted, however, the understanding of long COVID has improved, with hypotheses that are in keeping with the findings of improvement after vaccine. There is a distinct immunological profile of long COVID sufferers, characterised by low levels of IgM and IgG3, which, in conjunction with other basic determinants such as previous history of asthma and some of the primary symptoms, accurately predicts risk. In this scenario, boosting the immune system may help rectify this deficiency [8].

There is also evidence of abnormalities in the gut microbiome at diagnosis, which is exaggerated over time with long COVID. In a study of 106 people with long COVID in Hong Kong, three-quarters had persistent symptoms at 6 months [28]. There were differences in the microbiome between people who develop long COVID and those who make a complete recovery from their COVID-19. This supports the hypothesis of persistent virus residing within immuno-privileged tissue, triggering persistent low-grade inflammation, with associated fatigue, increased thrombosis risk, and cognitive impairment, as an underlying cause of the symptoms. In this case, the boost to the immune system of the vaccine may help the eradication, or at least suppression, of a residual viral load.

These results suggest that, for the majority of long COVID patients, receiving a vaccination not only offers protection against reinfection from subsequent variants of COVID, but also potentially triggers improvement in symptoms. Clearly, the observational nature of these data cannot be used to imply causation, and it would be inappropriate to consider mechanisms at this point. These data need verification with a prospective randomised controlled trial.

## 8. Limitations

It is important to acknowledge that there is no time control in this survey, and the natural history of long COVID is uncertain. As such, from these data it is impossible to determine whether each vaccine offers some benefit, with an additional benefit for those receiving the Moderna vaccine, whether those receiving the AstraZeneca/Oxford vaccine represent a “time control” with the Moderna vaccine demonstrating a significant independent benefit, or, for that matter, whether this latter group represents the time control, and the adenoviral vector vaccines exacerbated symptoms. This latter alternative is very unlikely considering the majority of patients had experienced symptoms for at least 9 months, and there was a return to the baseline for at least half of the patients within a month of vaccination.

The online survey recruited participants predominantly via social media and is unlikely to be representative of the population of people with long COVID. Most respondents identified as white (90.8%) and just over 80% were female, which in the case of the latter is a much higher proportion than that reported by the ONS [6]. The age range of respondents was broader, with a good representation between the ages of 31 and 65, and a further 65 respondents over 65, but only three respondents under the age of 20. The survey asks respondents to report their current symptoms and recall their symptoms pre-vaccination, with some individuals having to remember how they were feeling several weeks beforehand, possibly resulting in recall bias. Although the numerical recall may be flawed, the overall trend in symptoms is likely to be robust. Specifically, an individual may not be able to accurately recall whether their score was a 7 or 8, however they are likely to accurately remember that their symptoms have improved rather than deteriorated.

Finally, as already mentioned, we were not able to include a control group of unvaccinated participants; however, over 80% of those who completed the survey had been suffering symptoms for more than six months and we therefore consider the probability that any recovery was spontaneous as fairly low.

## 9. Conclusions

From a sample of 812 participants with long COVID, the majority of patients reported their symptoms had improved after their first vaccination and only a small number experienced a deterioration. These symptoms returned to baseline in over half of participants, suggesting that this was, at least in part, a direct vaccine effect. This requires validation in prospective randomised controlled trials.

## Figures and Tables

**Figure 1 vaccines-10-00652-f001:**
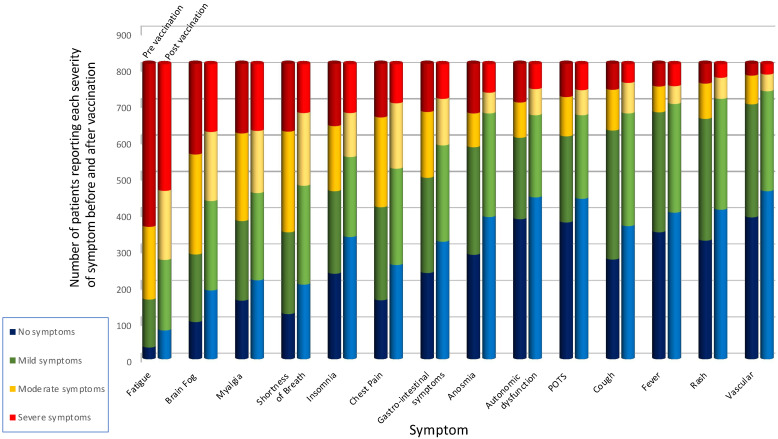
Proportion of patients with no, mild, moderate and severe symptoms before and after vaccine. For each symptom, pre-vaccination status is to the left, post vaccination status is to the right. No symptoms, blue, score 0/10; mild symptoms, green, score 1–4/10; moderate symptoms, yellow, score 5–7/10; severe symptoms, red, score 8–10/10; POTS, Postural Orthostatic Tachycardia Syndrome.

**Table 1 vaccines-10-00652-t001:** Baseline characteristics of participants.

Characteristic	Total Population	Oxford AZ	Pfizer	Moderna
n	812	401	338	73
Sex (% Female)	80.6	78.7	83.7	76.7
Age Group (% in each group)				
≤20 years	0.4	0.2	0.6	0
21–30 years	3.7	2.6	5.3	2.7
31–40 years	18.2	15.4	19.8	27.4
41–50 years	29.6	27.3	32.3	30.2
51–60 years	32.7	37.4	28.4	24.3
61–70 years	13.0	15.2	10.6	13.7
≥71 years	2.5	1.8	3.6	1.4
Severity of Acute COVID (%)				
No symptoms	1.1	1.2	1.2	0
Mild symptoms	12.8	11.9	13.6	13.7
Moderate symptoms	75.2	76.4	74.3	72.6
Short Hospital stay	7.4	7.3	7.4	8.2
Longer Hospital stay ± ITU	3.6	3.3	3.6	5.5
Diagnosis of COVID				
PCR test	31.1	23.1	40.8	32.9
Antibody test	11.2	8.6	13.0	12.3
Symptoms & contact	8.9	7.2	9.8	15.1
Symptoms alone	46.8	58.4	34.9	34.3
Duration of long COVID (%)				
4–12 weeks	5.4	6.5	4.2	4.1
3–6 months	15.0	12.4	17.8	17.8
6–9 months	8.0	5.6	9.2	16.4
>9 months	71.6	75.5	68.9	61.6

**Table 2 vaccines-10-00652-t002:** Severity of primary symptoms (presented as absolute numbers (%)) (scored on a visual analogue scale from 0–10 with 0 = no symptoms and 10 = most severe symptoms—mild score 1–4, moderate score 5–7 and severe score 8–10).

Key Symptom	No Symptoms	Mild	Moderate	Severe
Fatigue	32 (3.9%)	132 (16.3%)	200 (24.3%)	448 (55.3%)
Brain Fog	102 (12.6%)	186 (23%)	275 (34%)	249 (30.7%)
Myalgia (Muscle Pain)	161 (19.8%)	219 (27.2%)	241 (29%)	191 (24.1%)
Shortness of Breath	124 (15.3%)	225 (27.9%)	277 (33.2%)	186 (21.9%)
Insomnia	235 (28.9%)	227 (28.0%)	179 (22.1%)	171 (21.1%)
Chest Pain	162 (20.0%)	256 (31.8%)	247 (29.7%)	147 (17.4%)
GI symptoms	237 (29.2%)	262 (31.3%)	181 (22.7%)	132 (16.9%)
Anosmia	287 (35.3%)	296 (36.0%)	93 (11.1%)	136 (15.9%)
Autonomic dysfunction	385 (47.4%)	224 (27.0%)	97 (11.8%)	106 (13.8%)
POTS	376 (46.3%)	237 (29.0%)	108 (13.2%)	91 (11.9%)
Persistent Cough	274 (33.7%)	355 (42.8%)	112 (13.9%)	71 (8.1%)
Fever	349 (43.0%)	330 (40.0%)	71 (9.0%)	62 (7.0%)
Rash (incl. COVID toes)	335 (41.3%)	326 (40.2%)	97 (12.1%)	54 (6.9%)
Vascular complications	390 (48.0%)	311 (37.7%)	79 (9.6%)	32 (3.8%)

POTS—Postural Orthostatic Tachycardia Syndrome. GI symptoms—Gastro-intestinal symptoms.

**Table 3 vaccines-10-00652-t003:** Percentage change in symptom score after 1st dose of vaccine (mean (95%CI)), stratified by vaccine administered after adjustment for baseline symptom score, age group, sex and duration of long COVID symptoms prior to vaccination in those with symptoms at baseline, and/or who developed symptoms as a result of vaccination. (A positive value represents an improvement in symptoms).

Symptom	AZ/Oxford	Pfizer	*p* vs. AZ	Moderna	*p* vs. AZ	*p* vs. Pfizer
Fatigue	13.7 (9.7–17.8)	17.8 (13.2–22.4)	0.4	26.5 (16.9–36.1)	0.009	0.08
Brain Fog	22.2 (17.4–27.0)	21.7 (16.1–27.2)	0.9	31.7 (20.6–42.9)	0.01	0.02
Myalgia	11.5 (6.1–16.8)	16.8 (10.4–23.1)	0.3	30.7 (18.3–43.0)	0.006	0.04
Shortness of Breath	23.2 (18.1–28.3)	24.5 (18.6–30.3)	0.5	33.7 (21.6–45.8)	0.07	0.2
Insomnia	23.6 (17.2–30.1)	28.5 (21.3–35.7)	0.8	30.2 (16.2–44.2)	0.1	0.2
Chest Pain/Palpitations	25.6 (20.0–31.1)	26.1 (19.8–32.3)	0.98	34.8 (22.1–47.5)	0.2	0.2
Gastro-intestinal symptoms	24.7 (18.5–31.0)	24.6 (17.4–31.8)	0.98	41.7 (27.8–55.6)	0.002	<0.001
Anosmia	33.5 (26.7–40.2)	31.5 (24.1–38.9)	0.7	46.3 (26.1–46.3)	0.4	0.2
Autonomic dysfunction	23.9 (16.2–31.7)	28.9 (13.6–44.3)	0.01	33.7 (25.4–41.9)	0.004	0.1
POTS	23.0 (15.3–30.7)	25.4 (17.0–33.7)	0.7	27.7 (11.7–43.7)	0.1	0.2
Persistent Cough	24.1 (16.4–31.8)	26.1 (17.1–35.1)	0.5	32.4 (15.5–49.3)	0.3	0.2
Fever	1.7 (−10.5–13.9)	22.4 (8.6–36.3)	0.4	36.2 (10.1–62.2)	0.1	0.1
Rash (including COVID toes)	29.7 (12.7–37.6)	32.4 (23.6–41.1)	0.7	33.9 (17.0–50.9)	0.6	0.6
Vascular complications	25.3 (15.7–34)	29.8 (19.1–40.4)	0.7	31.4 (10.8–52.0)	0.8	0.6

POTS—Postural Orthostatic Tachycardia Syndrome.

## Data Availability

Anonymised data is available from the corresponding author on reasonable request.

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
