# Peer review of "The Impact of COVID Vaccination on Symptoms of Long COVID: An International Survey of People with Lived Experience of Long COVID"

_vaccines, 2022, doi:10.3390/vaccines10050652_

Round 1

Reviewer 1 Report

This paper covers a relevant topic. This reviewer has identified several points which must be addressed, because current data are not real and are extremely biased.

The use of “very common” is suggestive considering the topic of long COVID. Authors should tone down these expressions. Use of web-based questionnaires, not confirming real COVID-19 diagnosis and bias of data collection should be clarified as limitations.

The introduction is completely out of date in relation to data. There are several meta-analyses reporting current long COVID data. A link or potential justification about the proposed hypotheses explaining long COVID and potential responses after vaccination is clearly needed for justification of this study. A clear aim and a hypothesis is clearly needed to be stated.

Methods: One week after vaccination is an extremely short period of time for any effect. This is a huge bias of the study and should be clarified in the text and the title. Why the survey was opened to “contact with a proven case or symptoms alone.”? Contact with a proven case does not represent nothing, this is a bias of selection. Just symptoms? Which symptoms? Like a flu? These criteria are highly heterogeneous. This reviewer suggests to conduct again the analysis with JUST those subjects with a real PCR diagnosis. Collecting hospitalization data from subjects and not from medical records another bias. No data about which post-COVID symptoms were collected, opened questions, follow-up from infection, etc.

The statistical analysis does not fit with the aim. Associations are not cluster; this is an incorrect understanding of the authors. No data on clustering is provided. No analysis after before and after vaccination data is provided.

Results: The first part is not related to the aim of the study. The second part described changes in the symptoms after vaccine but it is slightly chaotic. No data about when the infection was (defining acute or chronic long COVID), when the vaccine was applied and so on.

Discussion: Extremely poor, no hypotheses are provided

Reference: Videos and newsletter can NOT be used as reference for a scientific study.

Author Response

We thank the reviewer for their comments and can apologise unreservedly for the delay in responding. Predominantly caused by the authors being infected with COVID in the most recent wave (and inadvertently creating a further author/patient) and the lead author being the lead clinician for both acute and long COVID in expanding a clinical service to its largest level since the start of the pandemic.

The use of “very common” is suggestive considering the topic of long COVID. Authors should tone down these expressions. Use of web-based questionnaires, not confirming real COVID-19 diagnosis and bias of data collection should be clarified as limitations.

We thank the reviewer for these comments. These expressions have been tempered, and the limitations of the survey have been highlighted. Further we have performed a sensitivity analysis limited to just those individuals who had a confirmatory test to ensure the generalisability of the results

The introduction is completely out of date in relation to data. There are several meta-analyses reporting current long COVID data. A link or potential justification about the proposed hypotheses explaining long COVID and potential responses after vaccination is clearly needed for justification of this study. A clear aim and a hypothesis is clearly needed to be stated.

The introduction has been updated as of 23rd March 2023 with numbers and the most recent hypotheses. This includes two suggested alternative findings of people with long COVID that could be addressed by vaccinations in a section in the discussion. We believe it would be disingenuous to suggest that this was a priori, however, as the original driver of the study was to address the concerns of the community that the vaccine would exacerbate their symptoms.

Methods: One week after vaccination is an extremely short period of time for any effect. This is a huge bias of the study and should be clarified in the text and the title.

One week was the minimum period permitted before entering the study, however the median time period of 9 weeks. These details have been included in the results section

Why the survey was opened to “contact with a proven case or symptoms alone.”? Contact with a proven case does not represent nothing, this is a bias of selection. Just symptoms? Which symptoms? Like a flu? These criteria are highly heterogeneous. This reviewer suggests to conduct again the analysis with JUST those subjects with a real PCR diagnosis.

In the early phase of the pandemic, widespread testing was not available. The median duration of long covid at time of vaccination was 9 months, which was regarded as a strength of the study, as any change in symptoms over a 4 week period would then be unlikely to be the natural history of the condition. Further, the UK definition of long COVID does not require a confirmatory test, but accepts a self reported symptomatic disease as evidence. given the lack of access to PCRs or antigen testing in the first 12 months of the pandemic. We accept the limitations of this, however have to accept that this is the accepted diagnostic criteria. As suggested, we have performed a sensitivity analysis looking purely at people with a confirmatory test, which has reassuringly demonstrated similar numerical response.

Collecting hospitalization data from subjects and not from medical records another bias. No data about which post-COVID symptoms were collected, opened questions, follow-up from infection, etc.

As above, this limitation is embedded within the diagnostic criteria for long COVID. Approximately 10% of people were hospitalised, with only 3.4% needing prolonged hospital stay with ventilatory support. However the access to hospital services, particularly at the height of either of the first two waves, was an inherent bias in its own right.

The statistical analysis does not fit with the aim. Associations are not cluster; this is an incorrect understanding of the authors. No data on clustering is provided. No analysis after before and after vaccination data is provided.

We accept the analysis of clustering was inadequately explained, and has now been better explained in other manuscripts, notably by Chris Brightling in the PHOSP COVID study. This section has now been dropped from the manuscript

Results: The first part is not related to the aim of the study.

As above, this has been dropped

The second part described changes in the symptoms after vaccine but it is slightly chaotic. No data about when the infection was (defining acute or chronic long COVID), when the vaccine was applied and so on.

The duration of long COVID is presented in the baseline characteristics. This is also a co-variate in the analyses presented in table 3, which is clarified in the legend. We did attempt to collate data on the gap between onset of symptoms and duration of long COVID, however this was very poorly completed in the questionnaire.

Discussion: Extremely poor, no hypotheses are provided

This has been rewritten and a section for hypothesis has been added.

Of relevance, we have also been explicit that this questionnaire was purely for hypothesis generation and is now being followed up with an interventional randomised controlled trial. The reviewers comments have informed the protocol for this definitive study.

Reference: Videos and newsletter can NOT be used as reference for a scientific study.

This has been corrected.

Reviewer 2 Report

Overall, this is an interesting paper. It seems t me to be hypothesis generating, rather than confirming a hypothesis, and should be worded accordingly.

Specifics:

Abstract

lines 20-21 - caution about what? Be specific. How do you know this is true? Please specify COVID vaccination in particular - here and also on line 23.

line 26 - collected, rather than collated.

line 29 -  phrasing does not make sense. How does an Office have symptomology? Does that Office keep reports of symptoms? Also - Where is this mentioned in the main manuscript?

line 30 - please do not include results, such as 57.9% and 17.9%, in the Abstract that do not appear in the main manuscript.

line 35 - these data do not "Demand" validation - that is too strong a conclusion. They suggest that a less flawed follow up would be useful. 

Summary -note that several sentences do not end in periods.

line 40 - remove "very". Also, what is your reference for this assertion?

lines 45- 47 - Untrue. Overstates the impact of your results. At most, it suggests the value of a less flawed follow up study. 

lines 58, 60 - do you want to capitalize long in long COVID?

Lines 63 - 65 - this claim of priority in naming is unnecessary and should be deleted. 

line 81 - who was concerned? We need references. 

Results

Figures 1a and 1b (line 162) - what populations do the four colors represent? Was this before or after COVID vaccination? 

Some kind of summary statistics, like in the Abstract ~ 30, would be useful.

Discussion

line 242 "there appeared to be a clear advantage" is contradictory - either there IS a clear advantage, or there appeared to be an advantage (but it was not "clear").

Concludsion

line 280 - There were 812, not 900, participants. They reported their symptoms, instead of felt. 

line 284. Stop this sentence after the word trials. The rest of the sentence is not true and is misleading.

Author Response

We thank the reviewer for their comments and can apologise unreservedly for the delay in responding. Predominantly caused by the authors being infected with COVID in the most recent wave (and inadvertently creating a further author/patient) and the lead author being the lead clinician for both acute and long COVID in expanding a clinical service to its largest level since the start of the pandemic.

Overall, this is an interesting paper. It seems t me to be hypothesis generating, rather than confirming a hypothesis, and should be worded accordingly.

We thank the reviewer for this comment, and have clarified this in several places. Further since the submission, we have commenced the randomised controlled trial to definitively answer this question, and we thank the reviewer for their suggestions to this manuscript which have also helped inform the trial.

Specifics:

Abstract

lines 20-21 - caution about what? Be specific. How do you know this is true? Please specify COVID vaccination in particular - here and also on line 23.

The drive to perform this survey came from the three patient support groups that are included as co-authors in this co-produced piece of work, Namely longCovidSOS, the ZeroCovid Alliance and the research Aid Network. As all three of these groups highlighted this as a concern, Ami and I accepted their opinion. We have clarified exactly what we mean

line 26 - collected, rather than collated.

This has been corrected

line 29 -  phrasing does not make sense. How does an Office have symptomology? Does that Office keep reports of symptoms? Also - Where is this mentioned in the main manuscript?

We apologise, for this assumption. The UK Office of National Statistics collates all of the self reported symptoms in the UK for everything from incident acute COVID, through long COVID symptoms and prevalence to the UK census data. The specific reference in the manuscript has been expanded on line 59 and is supported by Daniel Ayoubkhani’s team. (reference 5)

line 30 - please do not include results, such as 57.9% and 17.9%, in the Abstract that do not appear in the main manuscript.

These numbers have been added to the main manuscript on line 185 and 188

line 35 - these data do not "Demand" validation - that is too strong a conclusion. They suggest that a less flawed follow up would be useful. 

 We fully appreciate this comment and have tempered the statement

Summary -note that several sentences do not end in periods.

line 40 - remove "very". Also, what is your reference for this assertion?

We have no reference for this assertion, these are the comments of those with live experience that make up half og the research team and represent three major long COVID support groups. Given the population that Ondine, Vicky, Lyth and Jeremy represent, I have accepted their

lines 58, 60 - do you want to capitalize long in long COVID?

The UK convention is to capitalise COVID as an acronym for CoronaVIrus Disease, with the exception of the support group names, that do not do so. I would be happy to have that changed in accordance with the journal style

Lines 63 - 65 - this claim of priority in naming is unnecessary and should be deleted. 

This has been removed

line 81 - who was concerned? We need references. 

As stated above this is very difficult to reference as this was brought to the NHS long COVID task force by the patient support groups for those with lived experience. The founders of two of those support groups and an advocate for another group are included as co-authors.

Results

Figures 1a and 1b (line 162) - what populations do the four colors represent? Was this before or after COVID vaccination? 

This figure has been deleted as they give a very crude account of the clusters which was novel at the time of writing however have since been surpassed by the work of Chris Brightling in the PHOP-COVID study

Some kind of summary statistics, like in the Abstract ~ 30, would be useful.

This has been added

Discussion

line 242 "there appeared to be a clear advantage" is contradictory - either there IS a clear advantage, or there appeared to be an advantage (but it was not "clear").

The word clear has been removed as this would be inappropriate in this hypothesis generating survey

Concludsion

line 280 - There were 812, not 900, participants. They reported their symptoms, instead of felt. 

This has been corrected

line 284. Stop this sentence after the word trials. The rest of the sentence is not true and is misleading.

As has this…

Thank you again for your comments

Round 2

Reviewer 1 Report

To be honest, authors have not addressed most comments. 

The introduction is again out of date. There is no meta-analysis pooling data on long COVID in the introduction. there is no data on long COVID. 

The  the UK definition of long COVID is out of date. The WHO publishes a new definition in December 2021, so authors should use this one. This topic is related to self-reported symptoms as a permitted diagnosis. 

The introduction again does not include any of the studies on vaccines and long COVID. there are several now. 

In conclusion, the introduction and discussion must be updated and again rewritten

Author Response

To be honest, authors have not addressed most comments. 

We are sorry you feel this way. We believed we had addressed each of the points raised in turn, with the exception of the inclusion of meta-analyses as to the symptoms. Our thoughts were that, as we were presenting a hypothesis generating survey rather than a review of the existing data, that presenting the lack of consistency in the existing data, with a range from 5-78% of patients whi had confirmed COVID-19 suffering from the condition highlights the need for further research.  We have modified the introduction to include the most recent meta-analysis of the symptoms of long COVID. Given that we are presenting a hypothesis for treatment, we have avoided considering mechanisms in the background.

The introduction is again out of date. There is no meta-analysis pooling data on long COVID in the introduction. there is no data on long COVID. The  the UK definition of long COVID is out of date. The WHO publishes a new definition in December 2021, so authors should use this one. This topic is related to self-reported symptoms as a permitted diagnosis.

We have updated the introduction to refer to the WHO definition, in addition to the UK definition. As the latter was the definition at the time of the survey, this reference we used for the definition of the recruitment. We have included the most recent meta-analysis on the symptoms of long COVID, and would happily include any specific analyses suggested by the reviewer as there are many, including one of our own, which we have deliberately avoided self-referencing  

The introduction again does not include any of the studies on vaccines and long COVID. there are several now. 

As none of these were available at the time of conducting the survey we have not included these in the introduction, however we have added several of these into the discussion. These include multiple smaller studies that have variably suggested no significant difference and paradoxically implausibly large differences. Our survey, however was twice the size of the nearest comparator survey, and up to 20 times larger than some of the previous reports. We want to stress, however that the size of the survey does not make it any more robust, just more compelling for further definitive investigation.

In conclusion, the introduction and discussion must be updated and again rewritten

These have been rewritten in order to update them with the latest evidence, within the word limits of the manuscript.

Reviewer 2 Report

Abstract has some awkward wording that needs editing (eg. Lines 21, 22-23). It is unclear what recommendations are being addressed on line 36; as with any Conclusion, this comment should match some sort of previously stated Aim of the project.

The authors seem to switch back and forth between long COVID (e.g., lines 56, 118) and Long COVID (e.g., lines 97, 115) – I suggest they pick either a small or capital letter L / l, and use that.

  1. Introduction

Not sure what ITU means (line 65). Similarly, what are CKD (line 66) and ONS (line 72) and NHS (line 92)?

Line 68 – A study does not perform evaluations, although authors may do so, or a study might describe detailed evaluations.

I continue to question assertions like in line 95. Who expressed these concerns to patient support groups? How do the authors know this is true? Surely you can figure out a way to reference this.   Similarly, who made the “suggestion” cited on line 98? If the authors cannot address these questions, I suggest they simply leave them out. They are not crucial to this paper.

  1. Results – Line 167-170 refer to a specific “ONS report” but there is no reference given.

I am unclear with Total population reported in Table 1 is 900, when the authors seem to be describing a smaller number of 812 as the size of the study population.

Table 2 needs a column for “None” =0 score, so readers can ensure that totals make sense. That would match the multi-coloured bars in Figure 1. Is Table 2 describing “before”, while Figure 1 is both before and after?

  1. Discussion – Without a reference, how do you know that there is a “concern expressed by many sufferers that long COVID was an autoimmune condition.”? (lines 258-259) Similarly, who had the hypothesis described on line 271?

Author Response

Abstract has some awkward wording that needs editing (eg. Lines 21, 22-23). It is unclear what recommendations are being addressed on line 36; as with any Conclusion, this comment should match some sort of previously stated Aim of the project.

This has been tightened up, and the final comment has been deleted.

The authors seem to switch back and forth between long COVID (e.g., lines 56, 118) and Long COVID (e.g., lines 97, 115) – I suggest they pick either a small or capital letter L / l, and use that.

This has been corrected and is now consistent throughout the manuscript with the exception of the references (which do switch regularly)

  1. Introduction

Not sure what ITU means (line 65). Similarly, what are CKD (line 66) and ONS (line 72) and NHS (line 92)?

Apologies the expansion for some of these abbreviations have been deleted in the last iteration, and others have appeared for the first time. ITU is intensive Therapy Unit, CKD – Chronic Kidney Disease, ONS - Office of National Statistics and NHS is the National Health Service. These have been expanded.

Line 68 – A study does not perform evaluations, although authors may do so, or a study might describe detailed evaluations.

This has been corrected

I continue to question assertions like in line 95. Who expressed these concerns to patient support groups? How do the authors know this is true? Surely you can figure out a way to reference this.   Similarly, who made the “suggestion” cited on line 98? If the authors cannot address these questions, I suggest they simply leave them out. They are not crucial to this paper.

These concerns were brought to us (Ami and myself, WDS) by the patient support group at the NHS long COVID taskforce meetings, the national organisation brought assembled by the UK government to explore treatment . As you say, it is very difficult to reference these concerns, we have therefore modified the wording significantly to “During the initial rollout of vaccines to protect against SARS-CoV-2 in early 2021, individuals expressed concern to patient support groups (via personal communication with OS, VvdT and LH) that vaccination may exacerbate any autoimmune disorder. In order to address this, we surveyed members across international long COVID support groups to determine the impact of vaccination on their symptoms.” 

  1. Results – Line 167-170 refer to a specific “ONS report” but there is no reference given.

Apologies. This refers to the earlier mentions Office of National Statistics report, reference 5. This has now been corrected

I am unclear with Total population reported in Table 1 is 900, when the authors seem to be describing a smaller number of 812 as the size of the study population.

Table 1 has been rewritten to just talk about those in the final analysis rather than including the people with ME and CFS that were excluded from the subsequent analysis

Table 2 needs a column for “None” =0 score, so readers can ensure that totals make sense. That would match the multi-coloured bars in Figure 1. Is Table 2 describing “before”, while Figure 1 is both before and after?

The extra column has been added as suggested. You are correct that the first column of Figure 1 represents the proportions of the absolute numbers laid out in table 2

  1. Discussion – Without a reference, how do you know that there is a “concern expressed by many sufferers that long COVID was an autoimmune condition.”? (lines 258-259) Similarly, who had the hypothesis described on line 271?

Apologies for not referencing this statment, there have been many reports about the autoimmune hypotheses. Although the most robust data comes from Danny Altman’s group at Imperial College, I have deliberately chosen one from a patient support groups to highlight the concern (Long COVID Kids).

The latter hypothesis is one that is currently being tested in our study at the moment. We have deleted this from the manuscript, and will hopefully be reporting the outcome of this trial in the near future.

Round 3

Reviewer 1 Report

The paper increases the quality each time. I do not agree with the authors sayng that when they did the study, those papers published in vaccine are not published. This is life and usually happens always. I think that introduction and discussion should include the current situation with evidence and the strenght of this paper, as the authors have defended in their responses. 

But, I do not want to back and force. This is an opinion. The Editor should take the decision, there is no problem

Author Response

Thank you for your comments and understanding. We have further updated the manuscript with some of the most recent publications from earlier this year. We hope to be following this manuscript up later in the year, having revisited the original participants and a wider group. Your comments have been incredibly helpful in preparation of that study